# Adaptive FPGA-Based Accelerators for Human–Robot Interaction in Indoor Environments

**DOI:** 10.3390/s24216986

**Published:** 2024-10-30

**Authors:** Mangali Sravanthi, Sravan Kumar Gunturi, Mangali Chinna Chinnaiah, Siew-Kei Lam, G. Divya Vani, Mudasar Basha, Narambhatla Janardhan, Dodde Hari Krishna, Sanjay Dubey

**Affiliations:** 1Department of Electronics and Communication Engineering, Koneru Lakshmaiah Education Foundation, Aziznagar, Hyderabad 500075, Telangana, India or sravanthi.engg@mriet.ac.in (M.S.); sravankumar.gunturi@gmail.com (S.K.G.); 2Department of Electronics and Communication Engineering, Malla Reddy Institute of Engineering and Technology, Maisammaguda, Hyderabad 500014, Telangana, India; 3Department of Electronics and Communications Engineering, B. V. Raju Institute of Technology, Medak, Narsapur 502313, Telangana, India; divyavani.g@bvrit.ac.in (G.D.V.); mudasar.basha@bvrit.ac.in (M.B.); harikrishna.dodde@bvrit.ac.in (D.H.K.); sanjay.dubey@bvrit.ac.in (S.D.); 4College of Computing and Data Science (CCDS), Nanyang Technological University, Singapore 639798, Singapore; siewkei_lam@pmail.ntu.edu.sg; 5Department of Mechanical Engineering, Chaitanya Bharati Institute of Technology, Gandipet, Hyderabad 500075, Telangana, India; njanardhan_mech@cbit.ac.in

**Keywords:** posture recognition, localization, FPGA, service robot, sensor fusion

## Abstract

This study addresses the challenges of human–robot interactions in real-time environments with adaptive field-programmable gate array (FPGA)-based accelerators. Predicting human posture in indoor environments in confined areas is a significant challenge for service robots. The proposed approach works on two levels: the estimation of human location and the robot’s intention to serve based on the human’s location at static and adaptive positions. This paper presents three methodologies to address these challenges: binary classification to analyze static and adaptive postures for human localization in indoor environments using the sensor fusion method, adaptive Simultaneous Localization and Mapping (SLAM) for the robot to deliver the task, and human–robot implicit communication. VLSI hardware schemes are developed for the proposed method. Initially, the control unit processes real-time sensor data through PIR sensors and multiple ultrasonic sensors to analyze the human posture. Subsequently, static and adaptive human posture data are communicated to the robot via Wi-Fi. Finally, the robot performs services for humans using an adaptive SLAM-based triangulation navigation method. The experimental validation was conducted in a hospital environment. The proposed algorithms were coded in Verilog HDL, simulated, and synthesized using VIVADO 2017.3. A Zed-board-based FPGA Xilinx board was used for experimental validation.

## 1. Introduction

Human–robotic interaction (HRI) systems have become increasingly integrated into healthcare environments [1] (Alzheimer elders assistance), playing a vital role in assisting with tasks such as patient monitoring, medication delivery, and physical support. These robots must not only interact with their surroundings but also adapt to dynamic changes in real time to provide effective assistance. One of the key challenges in developing such systems is the precise localization of a human subject at stationary and moving positions [2]. Reliable and accurate human localization is essential for enabling robots to provide timely and context-aware responses and enhance the patient care and safety. Over the past 30 years, significant research has been conducted on human posture, sleep analysis, and sleep monitoring. In HRI systems, the early stage is about sensing and analyzing human activity recognition to serve as a better way of interacting with autonomous attendants. In this regard, human activities are confined to the sleep and activity stages.

According to the authors of a review analysis [3], sleep apnea affects between 9% and 38% of the general population, and this number is expected to increase in the future. Services navigate towards the localization point of elders/people with Alzheimer’s. Human localization has been classified into three categories: monitor-based localization (MBL), device-based localization (DBL), and proximity-based localization. The proposed HRI system is driven by ultrasound MBL algorithms [4]. In the HRI system, the next stage is service robot-based Simultaneous Localization and Mapping (SLAM) navigation. Recent market trends in statistical analysis shows that the global robotic healthcare AI market is expected to reach INR 188 billion by 2030, increasing CAGR by 37% from 2022 to 2030 [5].

In an HRI system, sensing has been performed in two ways by humans and autonomous robots. In this process, estimating the human pose in the environment plays a critical role in enabling the robot to assist humans in delivering necessary services. Human localization analysis using various sensing methods has been presented over the last two decades by researchers. The main challenge involved in pose estimation analysis relies on the quality of sensor data acquisition and pre-processing. Researchers have used pressure, non-contact, wearable, and non-wearable sensing devices to collect data [6,7,8]. The author focuses on in-bed human pose estimation, including sleep and sitting positions, using a multimodal conditional variational auto encoder (MC-VAE) and HRNet for single-modality inference [9,10]. A novel body posture recognition system on a bed, which accurately estimates sleep postures (supine, left lateral, prone, right lateral) using ballistocardiogram signals, enhancing comfort and reliability, was presented by the authors of [11]. The author of [12] focused on predicting sleep postures, including sitting using a Bayesian network algorithm with a heartbeat rate and image monitoring for accurate posture recognition in wireless body area networks. Similarly, a human sleep posture recognition method using millimeter-wave radar and interactive learning to overcome the sensitivity of radar signals to different individuals was presented by the authors of [13]. Wearable and non-wearable sensors, including three-axis accelerometers, multi-modal sensor fusion, electroencephalography (EEG), and thermometers, have been used to capture data on sleep posture [14,15]. The proposed system was developed using ultrasonic sensor fusion data and a contactless sensing approach to estimate human poses in the environment.

For autonomous robot navigation, SLAM approaches have been used. A robotic assistance system can provide care and support for humans, including those localized on a bed, thereby demonstrating the potential of robots to collaborate in various care scenarios [16]. Ultrasonic sensor fusion data are provided to the robot for effective navigation, allowing it to accurately detect human positions and assess distances, thus enabling smoother and more efficient movement through its environment. The authors have developed robotic delivery systems such as MEDROBO, which can enhance patient care by automating medicine delivery and monitoring vital signs for bedridden individuals using RFID tags attached to the bed [17]. A multifunctional intelligent bed (MIB) integrates autonomous movement, position adjustment, and interaction interfaces to assist mobility-impaired individuals, showing the potential for robotic assistance for bed-localized humans [18]. The challenges involved in the above approaches include maintaining accurate optimization and adaptability, particularly in adaptive healthcare environments. Issues such as RFID tag misplacement and the need for continuous real-time sensing can affect its efficiency in automating medicine delivery and assisting individuals with mobility impairment. Therefore, there is a need for IoT-based systems that enhance healthcare robotics by enabling real-time data collection and communication, thereby improving the task automation and accuracy. IoT-based robotic medicine delivery systems for bedridden individuals, improving outcomes, and overall healthcare efficiency in hospitals, have been addressed by the authors of [19]. Hu. Q. et al. proposed a system that employed a pressure sensor array integrated into a bed sheet with 1024 nodes for comprehensive data collection [20]. The author focused on a mobile robot using ZigBee for bed localization and drug identification based on a central processing unit (CPU) [21]. The key literature findings are that human localization and robot navigation progress independently in the best way. The major challenge is dynamic human localization [2] versus robot services [22], which has been addressed by a few researchers. SLAM methods have been addressed in the last four decades, and the navigation algorithm [23,24] is a subset of the data structure, as well as the shortest path algorithm, tree algorithm, and graph models such as Dijkstra, A* algorithms, and heuristic approaches. Bresson et al. [25] proposed an autonomous navigation method. A challenge mentioned is that navigation methods for person-following mobile robots are required in services. This challenge motivated the proposed research, and recent studies have developed a person-following approach using computer vision methods [26].

The HRI final stage has been accomplished with computational devices, and active research has been conducted using edge computation devices such as the Central Processing Unit (CPU), Graphical Processing Unit (GPU), and field programmable gate array (FPGA). Processing large amounts of data with a high computational speed is not sufficient with a CPU. An FPGA offers parallel data processing with lower latency and power consumption, making it ideal for real-time IoT features such as real-time sensor data collection, processing, and communication for efficient and adaptive systems. The authors of [27] discussed an FPGA-based smart delivery bot for goods, not medicine, utilizing sensors and the Dijkstra algorithm for efficient navigation. The authors developed an Internet-of-Things controlled robot using FPGA, enabling remote navigation via voice commands from a mobile app, with sensor data uploaded to the cloud for task completion [28]. The authors presented FPGA-based robotic accelerators as competitive alternatives to CPU and GPU platforms [29,30], focusing on their performance and energy efficiency, while analyzing optimization techniques and technical challenges within the robotic system pipeline, including commercial and space applications [31]. By leveraging FPGA architectures, the authors proposed an adaptive FPGA-based accelerator for human–robot interactions in indoor environments.

Utilizing an FPGA as a hardware accelerator represents a significant advancement, greatly enhancing computational speed and parallel processing capabilities. The emphasis of this algorithm on FPGA technology makes it particularly suitable for real-time application. The key contributions of the proposed approach are as follows:Novel Hardware Schemes are presented for static and adaptive human localization analyses in indoor environments.The proposed accelerator is a novel heuristic-triangulation-based navigation algorithm for achieving adaptive SLAM.FPGA-based accelerators are proposed for establishing interactions between human localization and robot systems.

The research presented in this paper includes the following components. Section 1 outlines the background and motivation behind human sleep posture analysis using the FPGA implementation. Section 2 and Section 3 details the proposed methodology, including both theoretical and hardware aspects. In Section 4, the proposed method is validated using results related to the synthesis, power consumption, and experimental comparisons. The final section summarizes the research findings and discusses future research directions.

## 2. Hardware-Based Algorithms

Human–service robot interactions are embedded with the localization of the human and robot and the robot provides services based on the navigation algorithm under event-driven conditions. Hardware-based algorithms were developed to analyze human localization in both static and adaptive scenarios. Based on human localization, the proposed hardware-based triangulation-navigation algorithm was developed using a service robot. Table 1 represents the related symbols and abbreviations used through out this research work.

### 2.1. Hardware-Based Algorithm for Human–Robot Interaction

Figure 1 presents an overall flowchart of the proposed hardware-based human–robot interaction (HRI). The proposed HRI methods depend on the localization of the subject (human) and robot. Human localization was performed using contactless sensing. Localization is associated with both static and adaptive forms. The same information is transmitted to the service robot. Before planning to serve the human, the service robot self-localizes based on direction and triangulation methods. After receiving the coordinates of the human localization, it navigates towards the destination using a triangulation-based navigation algorithm. After the successful accomplishment of the task, it retrieves to its parking station using the same navigation algorithm.

#### 2.1.1. Hardware-Based Algorithm for Human Localization in an Indoor Environment

This portion focuses on determining human localization within an indoor environment.

The pseudocode for locating humans in various scenarios is outlined in Algorithm 1. The explanation of the pseudocode in Algorithm 1 is represented in the form of a flowchart, as shown in Figure 2.**Algorithm 1:** Pseudo code for Hardware-Based human localization  Initialize sensory fusion distance data  always @ (posedge clk) begin  {PIR, Human available, Position values, Two Positions, Sleep posture, sitting posture, one hot} = 0.  state = INIT.  Case (state)  State_1: (PIR = 1)? Human available: state.   Case (Human available)   State_11: (Position values > Two Positions) Sleep posture: Sitting posture.    Case (Sleep posture)    State_1a: (Posture = 6′b111111)? Supine: State_1b.    State_1b: (Posture = 6′bxx0101)? Right side posture: Left side posture.    Case (Sitting posture)     State_1as: (((Position && Time) = Static)? State static: Adaptive.      Case (State static)        State_a1: (Position = one hot)? State_a2: Aliasing_pose.        State_a2: (Position = 6′dx)? Position Binary Classifier: state.      Case (Aliasing_pose)        State_b1: (Position = 6′dy)? Position Binary Classifier: state.     Case (Adaptive)        State_b11: if (count && CP == end position) begin            count && CP <= start position.             else {count <= count + 1, CP <= CP + 1}, {PP_n = CP}.        State_b12: (count = 0)? State static: State_b14.        State_b13: (CP = PP)? State static: State_b14.        State_b14: (CP = PP_n)? Position Binary Classifier: State b15.  default: state = INIT.  end case, End.

Algorithm 1 describes the human localization process using a hardware-based algorithm. As shown in Figure 1, sensor data were acquired using distance and PIR sensors (line 1). All parameters were initialized with a reset and operated based on clk (lines 2 and 3). The proposed human-localization algorithm was executed when the PIR sensor was evaluated for human availability. Distance sensor fusion was arranged in the form of six-bit posture values. When the position values were more than two, the sleep posture was considered (line 8). Sleep postures were classified in simple forms, such as supine and left- and right-side sleep postures (lines 9 to 11). In addition to the sleeping posture, the sitting posture is considered by the proposed algorithm (lines 12 to 18).

The sitting pose was evaluated based on the sensor distance, and its position was vital to the service robot to analyze whether the subject is in a sitting posture at a static position; if the subject traverses from one position to another, it is considered as an adaptive sitting position, which has been evaluated based on the time and position parameters (line 13). Adaptive positions were evaluated using lines 19–25. The static pose positions depend on the coverage of the sensor fusion. The sensor coverage of sitting positions includes complete individual sensors and aliasing situations (sensing of more than one human sensor). Digital one-hot-based sensor fusion was evaluated for the subject position among the six positions on the bed. Positions were classified using a binary classifier (line 16). In the aliasing situation, sensor fusion coverage with an encoded one-hot approach was embedded and classified using a position binary classifier (line 18).

The hardware-based adaptive sitting position analysis is the first of its kind to provide better services using robots. Thus, the proposed method provides superior robotic services. Count, past position (PP), and current position (CP) are parameters for adaptive sitting position evaluation. The start position was initialized by sensors, and every step of the movement from one position to another position was recorded until the subject stopped moving (lines 20 to 22). Each stage’s past position was memorized as PP_n, and the proportional adaptive time was registered as the count value. Adaptive current positioning was then classified and shared with the updated new position of the service robot (line 25). Table 2 presents the classification of the positions in the lines into a binary classifier using sensor fusion data.

#### 2.1.2. Hardware-Based Adaptive SLAM

The proposed hardware-based adaptive Simultaneous Localization and Mapping (SLAM) was used to serve humans in an indoor environment. In robotics, localization and mapping are embedded in navigation systems. In the generic approach, the robot navigates to the destination early using robot navigation. In this process, SLAM plays a vital role in allowing the robot to continuously update its localization. Based on the obtained path, it plans to retain its route by mapping. When serving a human (subject), the subject changes from one portion of the bed to another. In this regard, the proposed adaptive SLAM approach was developed to serve adaptive situations according to human movement.

The proposed method was developed as a heuristic-type triangulation-based navigation algorithm to achieve an adaptive SLAM.

Figure 3 shows a service-based robot in an indoor environment, initially a robot at a parking station. As mentioned in Algorithm 2, the robot executes services based on task assignment. Standard services have been used during the COVID pandemic, such as service robots serving subjects (humans) in isolation/individual. Standard tasks include food and medical services along with standard timings. Simultaneously, the robot registers the tasks of either food or medical services. As presented in Algorithm 2, the robot is assigned to either a multitask or a single task in line 4. In the case of the multitask, the sorting of the path is performed using the bubble sort technique, and the robot traverses to the task point (line 7). SLAM retains its localization and is mapped in line with triangulation-based navigation. The task coordinates were registered and utilized until the task was completed. The path plan mapping was computed using the angles of the robot localization coordinates and task coordinates (lines 9–13).

Adaptive SLAM was established once the robot received the delivery point (task destination) in the traversing mode. The robot adapts its path based on the adaptive movements of the subjects. If the subject moves to a new location, the robot can adapt the path and travel to deliver the item (lines 14 to 17). The explanation of Algorithm 2 is represented in the form of a flowchart, as shown in Figure 4. The use of triangulation navigation is shown in the lines presented in Figure 5. Triangulation navigation supports the performance of static and adaptive SLAM, as indicated by Algorithm 2. Figure 5 shows the subject in bed at various locations. Paths 3a and 3b are the shortest paths among the path distance weights when the robot navigates. Similarly, the longest distances for the robot navigating from the gateway to the destination of the task were paths 6a and 6b. The robot moves back using the same route that it has taken to accomplish the task until the endpoint of the gateway (i.e., the entrance gateway). It then transitions to the parking station if no other tasks are scheduled. If any task is assigned, Algorithm 2 follows.**Algorithm 2:** Pseudo code of hardware based Adaptive SLAM for robotic services  Service Robot (S_R_) initialize services and ready at parking station.  If (Task assigned)  Case (Task type)  State_1: (Task type = Multitask)? State_2: Static SLAM.  State_2: (Task sort)? Task destination: Task sort.  Case (Task_destination)  State_11: (Task_destination = Static Task sort _n)? State_12: Accomplished.  State_12: (Task_destination = Adaptive Task)? Adaptive SLAM: SLAM static.  Case (Static SLAM)   State_SS1: (Navigation sort_n = Task_destination_n)? Accomplished: State_SS2.   State_SS2: if (Current Node == Dest_ Node) begin            Current Node <= start position.               else {Current Node <= Current Node + 1}.  Case (Adaptive SLAM)   State_AS1: (Adaptive task)? Task assigned <= Adaptive task: Task_destination.   State_AS2: (Adaptive task sort)? State_AS3: State_AS2.   State_AS3: (Task destination <= Task sort)? Static SLAM: State_AS3.   end case, End.

## 3. Hardware Schemes for Human–Robot Interaction in an Indoor Environment

Section 3 deals with FPGA-based accelerators that are equivalent to accomplishing the proposed hardware-based algorithms for human–robot interactions. Concurrently, human and robot localization has been confined to different reconfigurable devices. The robot continuously receives static or adaptive human localization because it navigates to the destination using the triangulation approach.

### 3.1. Hardware Accelerator for Human–Robot Interaction

The proposed hardware accelerator is versatile for human–service robot interactions, as illustrated in Figure 6. It was integrated with human localization, robot localization, and navigation. In this research, two FPGA reconfigurable edge computing devices were incorporated; the first of these was for the human, and the other was for the robot. The pose control unit (PCU) is a part of the FPGA position for evaluating human localization. This is performed using the lines in Algorithm 1. Initially, the algorithm was triggered based on the PIR sensor information. Human availability was evaluated using PIR, which enables human localization computations. Six ultrasonic sensors {SH_R, SH_L, SA_R, SA_L, SR_L, and SL_L} were positioned on the roof to capture the human localization on the bed. These sensors were triggered by the PCU, and echo signals were captured and digitized at a distance from the sensor-distance fusion module. It is compiled with all the 20-bit sensor distances in the fusion with respect to the event conditions. Human localization was performed with five Processing Elements (PEs): sleep posture, static sitting position, adaptive sitting position, aliasing sitting position, and position binary classifier. The PCU verifies the classification of the subject during sleep or in the sitting position. Under these conditions, humans are versatile in the sitting position. If the subject is at one position until the robot begins its services, it is considered as a static position with six-bit information. Under certain conditions, the subject is positioned between two or three sensor coverage areas, and the subject is considered to be in an aliasing sitting position. This was addressed based on the maximum position zone. The adaptive sitting position evaluates the change in position by the participant with respect to time. All positions were evaluated using a position binary classifier and shared with the other end-edge computing device for human localization using ESP8266. It is operated at 9600 baud rates with the UART protocol to transmit the human localization details continuously.

An FPGA reconfigurable edge computing device was placed on the robot. It operates according to Algorithm 2. Four ultrasonic sensors {SF, SL, SR, and SB} were used on board to sense the environment. A Robot Control Unit (RCU) plays a vital role under various conditions. It triggers all the sensors and the communication module ESP8266. The proposed service robot was assigned a task based on its timestamp. For example, a robot can serve diabetic tablets based on the time at which the human wakes up from sleep (evaluated based on human movements). All services are registered in DDR3 memory. The AXI lite protocol was used to drive the data from memory. The pickup nodes were registered as tasks. Task sorting is in the sleep module for a regular navigation path until the human is positioned at a new location on the bed, compared to the previous destination task. Task-sorting PE has been revamping route nodes with mapping as per the human localization PE. Rerouted mapping nodes were provided to the navigation module. Robot localization is another parallel task that self-evaluates an edge device by using sensory information. Navigation develops a route according to the task nodes and robot localization using triangulation-based navigation approaches. Simultaneous Localization and Mapping (SLAM) was performed with 20-bit sensor distances and their fusion along with continuous destination task estimation.

### 3.2. Hardware Schemes for Human Localization

Figure 7 presents the detailed human localization information with internal architectures. The ultrasonic sensors were operated at a 40 KHz frequency and captured at a rate of 1/8 s. Distance data fusion was captured and stored in FIFO {H_R to L_L}. Prior to normalizing the six sensors’ distance as 6′bxxxxxx, it had been decoded as in the sleep position. For Sitting position at different levels, the sensory digital value is given as ‘1’ when it is in the availability range. Otherwise, this value is ‘0’. The sampling data are presented in Table 2. These data were used to classify either sleep or sitting postures using a pose-selection encoder. Sleep postures have been presented. There are more than two positions and the sensor data logic value is ‘1’. Then, it enables the sleep posture PE. Sensor data have been compared with the reference FIFO data on sleep posture, including supine posture (SP), left-side posture (LP), and right-side posture (RP), and they are encoded and transferred to the position binary classifier.

A pose-period encoder distinguishes between static and adaptive position modules. The static position has been presented in two ways: the static sitting pose PE and aliasing sitting pose PE. In the static sitting position PE, real-time sensor data were compared with the reference FIFO position. Similarly, aliasing and adaptive approaches have been used. To replace the sitting PE, an encoder was designed along the logic computation module. The logic computation module normalizes the aliasing coverage data with the adjacent higher grouping logic ‘1’. e.g., when HR = 1, HL = 1, and AR = 1 is 6′b111000, the subject is sitting in the cross mode between the left and right sides of the bed. Normalization confines the subject’s position to the right side of the bed. The adaptive position was confined to the pose period and the array of sensor data were compared. For example, if the subject initially sat at the HR, the sensor data weere presented as 6′b100000 at time T. It compares the subject’s movement from one position to another over time. The same is true for the PBC module. In this case, T + 1 with 6′b001000 and T + 2 with 6′b000010 indicate moving around the right side of the bed. Finally, the human was positioned at the right-limb coverage spot (RL). The relative information is shared with a binary classifier, and human localization is provided to the robot through IoT communication. The position binary classifier has been continuously evaluated in lines [32,33] of the binary search.

### 3.3. Hardware Schemes of Robot Localization and Triangulation-Based Navigation

Figure 8 shows the internal hardware schemes for robot localization and navigation. 

The real time sensor data of robot localization is represented in red color lines where as human localization is represented in blue color. The processing of data inside the architecture is represented in different colors. The environments were sectioned into sectors (sec_1… sec_n). Each sector was divided into multilevel sectors, e.g., sec_4 into sec_4_1 to sec_4_n. Robot localization was performed by comparing the real-time sensor fusion data and the location of the reference sector (s _1 to s _4_5). Human localization PE provides the destination of the human position, which is used for the assigned task. Every node has been considered a destination, and all sub-destinations (Des_1… Des_4_5) are routed to the end destination (Des-end), which has been assigned as the task. Concurrently, the time-stamp task assignment is aligned with human localization PE as a task assignment. The assigned tasks were sorted according to bubble sorting techniques. A task-based bubble sorting technique was constructed using eight register arrays and shifters. Four clock cycles were used to obtain the task-based sorted information. The information was shared with the mapping control unit to develop the route maps. Route maps were established based on triangulation-based navigation using a CORDIC IP core. The robot localization output is driven into the CORDIC and Euclidean distance computation modules. Triangulation computation inherent modules are CORDIC, which have been computed with 32 bits for the angles and their square root of distance calculation. Based on the task sort, information route maps were registered as static routes and adapted for routing in SLAM lines. Euclidean distance computation is performed with the two-bit navigation control and it estimates the distance with 32 bits between two nodes. The same procedure was performed by mapping the control unit to update the route maps. As shown in Figure 5, CORDIC and Euclidean provide an accurate distance to travel by the robot. The longest route from the gateway to the destination in the service sector is on the right side of the bed at the HR sensor-coverage node. Concurrently, an on-board reconfigurable computing device provides distances as paths 4a and 4b. The travel distance was evaluated using an odometer-based soft code as part of the execution unit. The execution controls the left and right motors of the robot with four bits each (two-bit acceleration control and two-bit direction).

## 4. Results

The proposed human–robot interaction was developed with FPGA-based accelerators. Human localization was evaluated at one edge of the FPGA device, and the robot on the other side on board was directed to the right destination (human positions) under both static, adaptive, and even driven conditions. The validation of the proposed algorithms is discussed in Section 2 and the hardware schemes are discussed in Section 3. The results were obtained with resource utilization at both ends along with power consumption. An experimental setup was established as a platform for validating the proposed algorithm in the form of experimental studies.

### 4.1. Resource Utilization

The proposed HRI utilizes Xilinx products from the Xilinx University program. The hardware accelerator proposed in Section 3 was developed with equivalent code by using Verilog HDL and was tailored as per the proposed HRI. The functional verification of the system was performed using the Xilinx simulator; Xilinx tools were used for the synthesis and implementation steps.

The reconfigurable devices are part of the Xilinx Zynq family, and their design has been mentioned by Xilinx (San Jose, CA, USA) as an XC7Z020-1CSG484 Zed board. The proposed approach utilizes the processing system (PS) to capture time-stamped tasks while executing the services. Control logic and interfacing were performed using the programming logic (PL) of the zed-board device. PS and PL were interfaced at the system level using the AXI lite protocol for synchronization at the complete system level. The Block RAM (BRAM) (140 blocks, 36 kb equal to 4.9 Mb) and 220 Digital Signal Processing (DSP) slices were on the zed board, which has been utilized to a certain extent to pursue this method.

Resources are consumed in the execution of human localization, as mentioned in Table 3, and robot localization and navigation are listed in Table 4. Among the computing methods, edge computation provides the fastest and most accurate method; FPGA reconfigurable devices consume less power and perform concurrently [34,35,36]. FPGAs for HRI solutions are the first to provide solutions under adaptive position–event conditions. Consumption in an FPGA was evaluated using lookup table (LUT), Block RAM (BRAM), and Digital Signal Processing (DSP) slices. Human localization consumed LUT (48%), BRAM (42%), and DSP slices (38%). Similarly, robot localization and navigation were used for LUT (57%), BRAM (46%), and DSP slices (39%).

Figure 9 and Figure 10 show the details of the consumption of each module as percentages. The adaptive sitting position PE for human localization and triangulation navigation PE for service robot implementation consumed the highest resource utilization. The architectures of service robot localization and navigation-based adaptive target positions were deployed in the FPGA and are presented in Figure 8. Table 3 and Figure 9 illustrate human localization resource utilization. Control units and interfacing implicit communication modules, such as UART, are consumed, as described above. The resource utilization of the other modules was as follows: sleep posture PE (11%, 10%, and 10%), position binary classifier PE (8%, 10%, and 12%), adaptive sitting position PE (25%, 31%, and 22%), and static and aliasing sitting positions PE (18%, 14%, and 20%). Similarly, Table 4 and Figure 10 show the device consumption as LUT, BRAM, and DSP slices of FPGA-based robots, such as the external modules and robot control unit (17%, 22%, and 25%), human localization PE (15%, 16%, and 14%), task assignment and sort PE (13%, 19%, and 12%), robot localization PE (17%, 12%, and 21%), triangulation navigation PE (24%, 22%, and 19%), and execution modules, UART, and display (14%, 10%, and 9%).

Figure 11 illustrates the power consumption of a human localization accelerator of 1.2 watts. The power consumption was registered using the Xilinx power estimation tool. The power consumption details of each hardware module are provided in percentage form in the pie chart in Figure 11. The adaptive sitting PE consumed 27% power; compared to other modules in human localization tasks, this value is high. The power consumption details of the other modules were as follows: pose control (23%), PE (18%), position binary classifier PE (12%), static and aliasing sitting PE (14%), and execution (6%).

Figure 12 shows the power consumption of the service-based robot localization and navigation accelerator. The overall device consumes 1.8 watts of power and presents the consumption of each module in percentage form, whereas the control unit and execution consume the same amount of power. The preferred tool is XPE, and the power consumption of each module is as follows: human localization (12%), task assignment and sorting (16%), data driven from the PS through the AXI protocol, robot localization (18%), and triangulation-based navigation PE (25%). Triangulation PE consumes more power than the other modules on the FPGA-based robots. The interfacing and control unit modules had the second highest share of power consumption.

### 4.2. Experimental Results

This section describes the experimental setup for human and robot interactions based on their localization under static and adaptive conditions.

#### 4.2.1. Experimental Setup

Figure 13 presents the experimental setup with two parts: one regarding the human posture and position analysis setup and another regarding the service-type FPGA-based robot. The human localization process is illustrated in Figure 13a,b. The service-based robot is shown in Figure 13c. A real-time experimental setup was used for the proposed hardware-based algorithms validated with the experimental setup in Figure 14a–d. Figure 14a shows the sensor coverage bed for capturing the human posture and positions. Figure 14b presents the human reverse supine sleep posture and the subject in the HL sensor coverage area in Figure 14c. The other subject was positioned in the RL sensor coverage area at the same time as the robot travelled to serve the subject.

Both experimental setups were embedded with ultrasonic sensors to capture human positions and explore the environment. The major experimental setups are sensing devices and other auxiliary devices such as a digital compass and PIR. The ultrasonic sensors were operated at 40 KHz and 5 volts, and the sensor array was operated using FPGA control units. A digital compass was deployed on the FPGA-based service robot to estimate the angles, which was also applied to CORDIC. Implicit communication was performed between the two devices using the ESP8266 Wi-Fi modules operated at a baud rate of 9600. The main unit in the proposed method is the FPGA. Xilinx-manufactured FPGAs are edge-computing devices, as mentioned in Section 4.1. The other parts of the robot experiment were embedded with two driven wheels and driving circuits, based on the control logic received from the triangulation-based navigation algorithm module. This is operated with driving circuits with LED acid batteries at 24 V/7 Amp. These batteries are charged every 6 h, and the operational time can be improved by replacing it with new battery technology. The FPGA and sensors are powered around 3.3–5 V from batteries through voltage regulators using 7805 IC modules. Figure 14c,d show the experimental details about the proposed hardware approaches. The robot frame was developed in in-house by local design experts. The robot frame has three layers: a bottom layer with batteries and voltage regulators, a middle layer with computational devices, and a layer on top of the service robot where service modules were positioned.

#### 4.2.2. Experimental Results of Human and Robot Interaction at Static Position

The FPGA-based robot services towards human localization at static positions are shown in Figure 15a–h. The versatile approach of the proposed method is that duo FPGAs are performed concurrently and communicate regularly through IoT modules. The FPGA-based robot initially triggered it to navigate from the parking point with respect to the timestamp task assignment, as illustrated in Figure 15a. Implicit communication has continued with the human localization analysis of FPGA accelerator devices.

The navigation was performed in lines of TNA to pick up the food/medicine drives towards sector 2, as presented in Figure 15b. The navigation in sector 2 continued until the pickup node was reached (Figure 15c) and the robot waits until the human positioned the food (PIR and ultrasonic sensor signal patterns were evaluated) on the robot (Figure 15d). The mapping module in Figure 8 provides a route map for navigating from sector 2 to sector 3 (gateway). Robotic turns are continuously evaluated by the CORDIC modules, and the distance to travel is provided by the Euclidean distance for successful navigation. Robot localization and self-positioning, along with kinematics and movements, are operated with robot control, and the positions are corrected based on the TNA with CORDIC modules.

The robot navigates from the gateway entry (Figure 15e) to sector 4 (Figure 15f) of the delivery bed and human localization. Based on human localization, a route map was mapped, and SLAM was performed using TNA to reach the destination point, as shown in Figure 15g. Similar to the pickup node, the human pattern identification of the subject indicated that food and medicine were collected. The proposed environment was considered ethical and not false or misleading in the collection of foods/medicines by the subjects. The FPGA-based robot returned from the delivery node to the parking station using a reroute with similar mapping techniques, as shown in Figure 15h. The return navigation by the FPGA adapts the mapping modules and TNA to reach the parking station. When the continuous task assignment continues to the robot, it is performed according to the assignments. In this approach, the delivery modes are inclined toward first-come, first-serve methods.

The static human-position-based services provided by the FPGA-based robot demonstration are as follows: https://www.youtube.com/watch?v=xIjKWfpmPIU (accessed on 25 September 2024).

#### 4.2.3. Experimental Results of Human and Robot Interaction at Adaptive Position

A demonstration of the adaptive-based position service by the FPGA-based robot is shown in Figure 16a–h. Similar to the previous experiment, it starts and collects the essential material from the pickup node (Figure 16a) and the navigates through gateway (Figure 16b). Adaptive human localization was initially observed at the HL sensor coverage area (Figure 16c). When the subject is in continuous motion, the human localization-side FPGA shares information with the FPGA-based robot. In this study, we considered coverage up to 50 cm from the bed. This coverage was provided to avoid false positives during the inference stage using the proposed method. The FPGA-based robot adapts to adaptive human localization, and proportional adaptive SLAM was performed (Figure 16e). In this demonstration, the subject reached the destination node as the RL coverage point (Figure 16f). The time duration was recorded to determine the rate of speed of the subject switching from one sensor to another, and the proportional total duration captured from the HL to RL nodes was 14.20 s. The robot moves from the delivery point to the parking station/next task point (Figure 16g,h). The experimental results are as follows: https://www.youtube.com/watch?v=cPasjy0F528&t=14s (accessed on 25 September 2024).

Table 5 presents a qualitative analysis comparing the contributions of various studies in this field of interest. Adaptive human localization-based services using autonomous robots have drawn attention owing to their effectiveness. In this regard, researchers are working on analyzing subject/human posture position groups [20,37,38,39]. Other robotics have been developed to serve subjects with different interaction methods, using face recognition and other methods. The integration of both ends has been attempted by very few researchers [40,41]. The proposed research contribution involves integrating both human localization and FPGA-based robot services; it is first of its kind to use adaptive-based robot services. FPGA edge computation provides better results than the other computation methods at the inference level [30].

The edge device clock frequency = 100 MHz, voltage applied for the device = 3.3 V, switching capacitance = 4.41 PF, dynamic power (P_dynamic) = 0.96 W, static power (P_static) = 0.24 W, and total power consumed for human localization is 1.2 W. The number of pipeline stages in hardware (S) is eight, clock time (Tclk) is 10 ns, and latency per iteration (S × Tclk) is 80 ns. The total number of iterations (N) was 30, and the total latency as (N + S − 1) × Tclk was 370 ns. The number of correct predictions was 29 and the total number of predictions was 30. The resulting accuracy was 98.4% and the error rate was 1.6%.
(1)$$Accuracy=\frac{No\;of\;Correct\;Predictions}{Total\;Prediction\;}\times 100=98.4\%$$
(2)$$Error=1-\frac{No\;of\;Correct\;Predictions}{Total\;Prediction\;}\times 100=1-\frac{29}{30\;}=1.6\%$$

The average response time for each path is represented as static in Figure 17 and adaptive in Figure 18. The time responses are the same up to the early delivery node, based on the Euclidean distance triangulation navigation towards various nodes around the bed. The return response time was the critical return time. The average response time of adaptive human–robot interaction (HRI) presents adaptive human localization and robot navigation. The adaptive response time changes when humans move from node1 to node4; its critical path and time response is 5 min 10 s.

## 5. Conclusions

In this article, we presented a service-based human–robot interaction system in healthcare environments. The proposed methodology successfully addresses a solution for human–robot interactions in delivering a task by analyzing adaptive human positions in a static or dynamic manner. FPGA-based accelerators have been developed as key solutions to address the challenges in healthcare assistance. Equivalent hardware accelerators were developed and deployed on a Xilinx FPGA XC7Z020-1CSG484 Zed board. The resource utilization of the proposed HRI for a human localization accelerator on a Zed board consumes 48%, 42%, and 38% for the LUT, BRAM, and DSP slices, respectively. Similarly, the robot localization and navigation consumed 57%, 46%, and 39% of the LUT, BRAM, and DSP slices, respectively. The device power consumption of the human localization accelerator is 1.2 watts and that of the service-based robot localization and navigation accelerator is 1.8 watts. The proposed human localization and triangulation navigation algorithm with parallel computing provided 98.4% accuracy compared with previous methods. In the future, the proposed method could be integrated with partial reconfigurations for multi-tasking services.

## Figures and Tables

**Figure 1 sensors-24-06986-f001:**
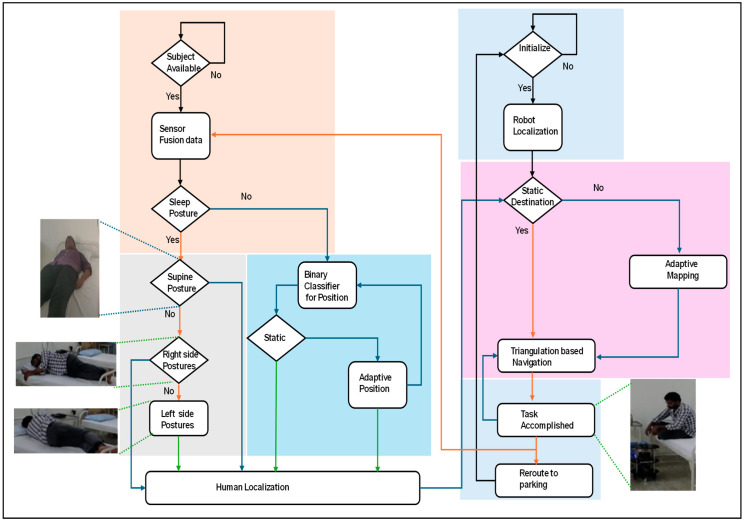
Flowchart of proposed hardware-based human–robot interaction.

**Figure 2 sensors-24-06986-f002:**
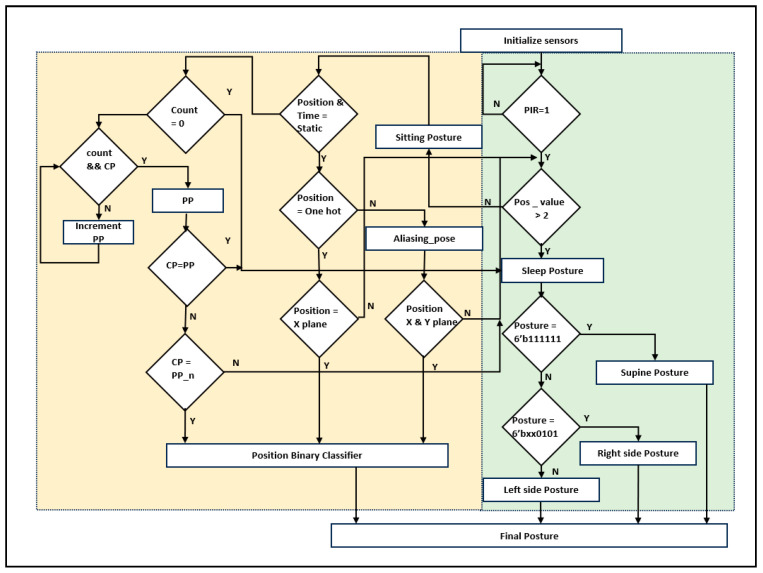
Flowchart for hardware-based human localization.

**Figure 3 sensors-24-06986-f003:**
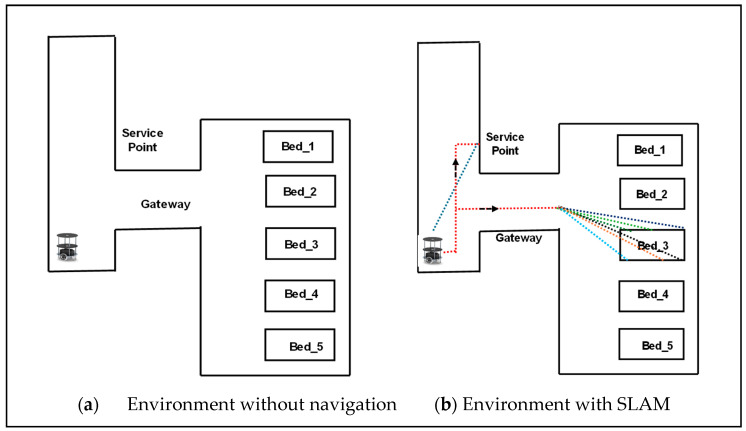
Triangulation-based navigation for service robots in an indoor environment. Different colored lines shows the representation of receiving signals from all sensors.

**Figure 4 sensors-24-06986-f004:**
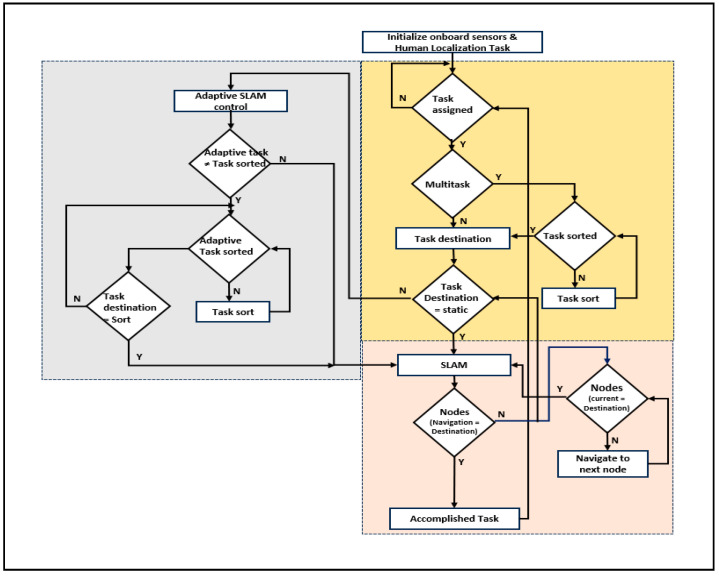
Flowchart for adaptive SLAM for robotic services.

**Figure 5 sensors-24-06986-f005:**
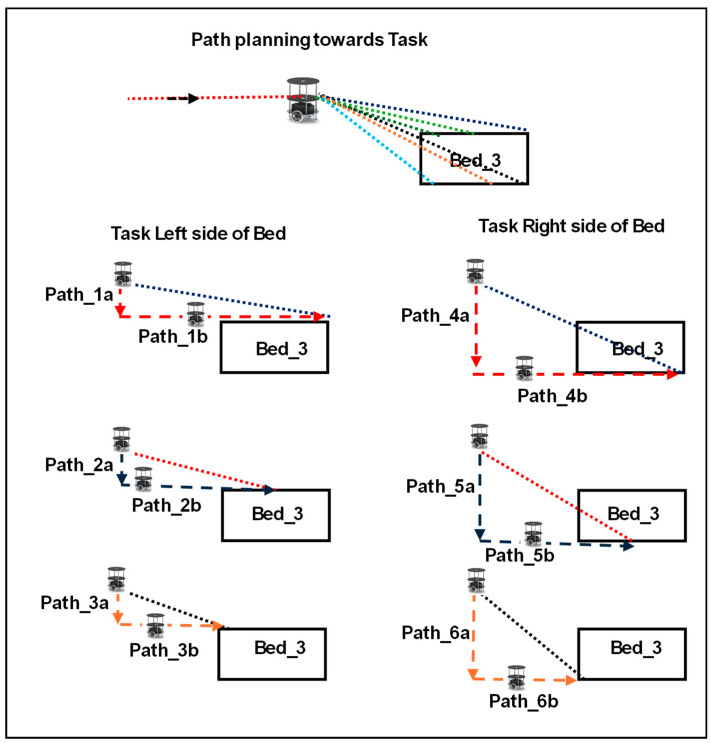
Path planning of service robots based on task in an indoor environment with different colors.

**Figure 6 sensors-24-06986-f006:**
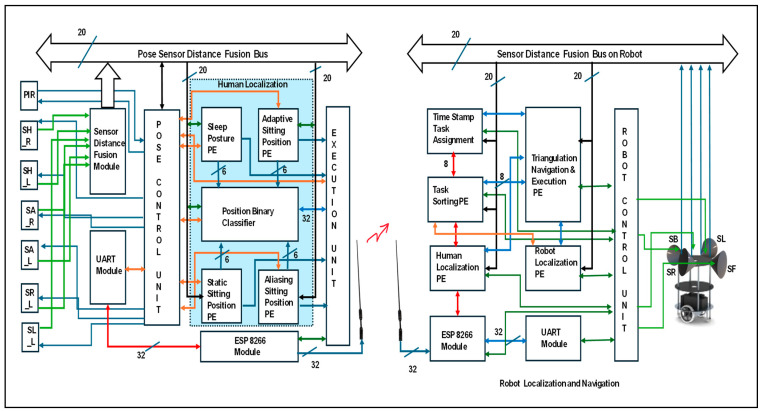
Overall hardware accelerator for human–robot interaction.

**Figure 7 sensors-24-06986-f007:**
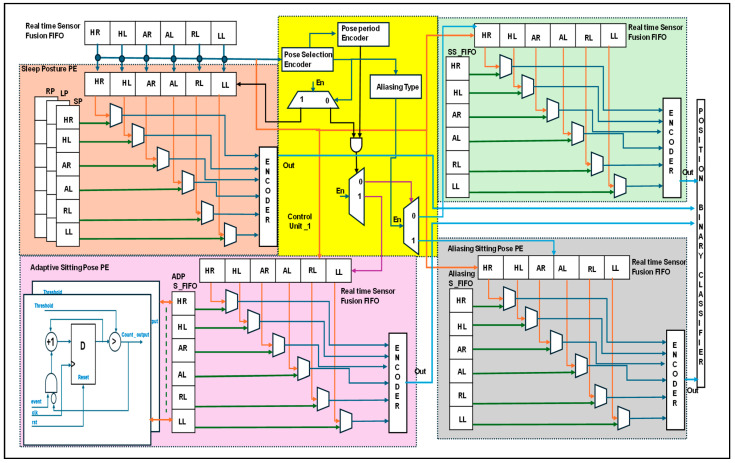
Internal architecture of the human localization process.

**Figure 8 sensors-24-06986-f008:**
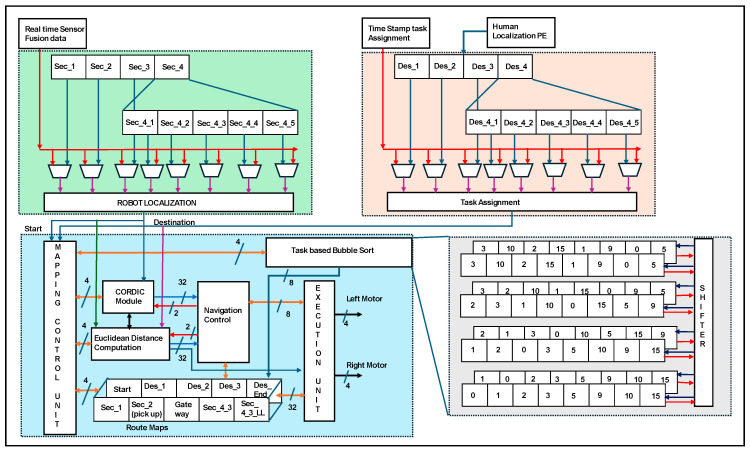
Hardware scheme robot localization and triangulation-based navigation.

**Figure 9 sensors-24-06986-f009:**
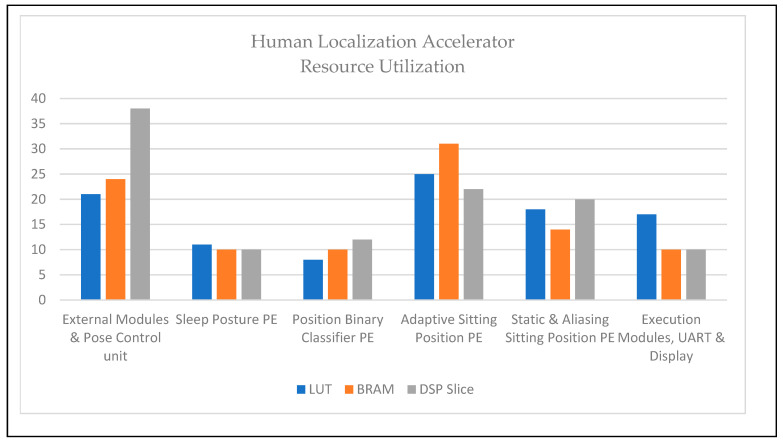
Resource utilization of human localization accelerator.

**Figure 10 sensors-24-06986-f010:**
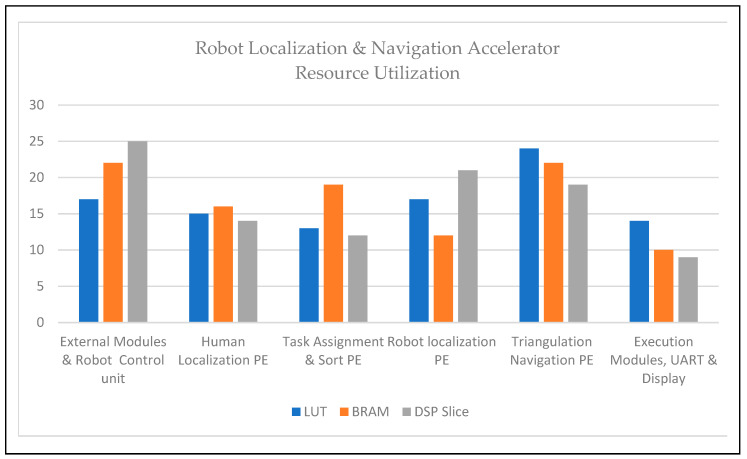
Resource utilization of robot localization and navigation accelerator.

**Figure 11 sensors-24-06986-f011:**
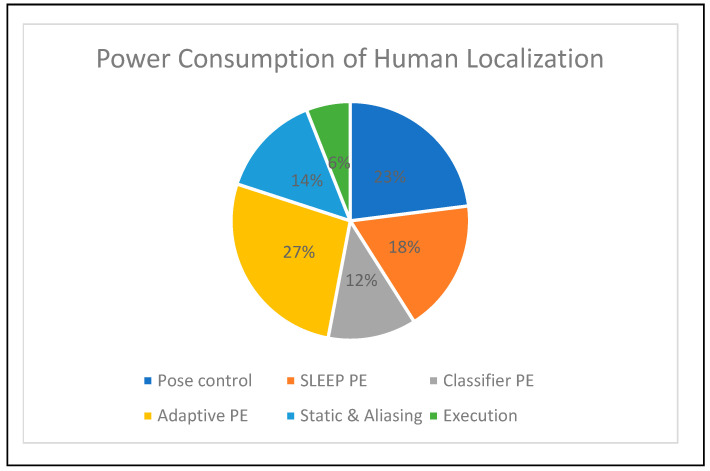
Device power consumption of human localization accelerator.

**Figure 12 sensors-24-06986-f012:**
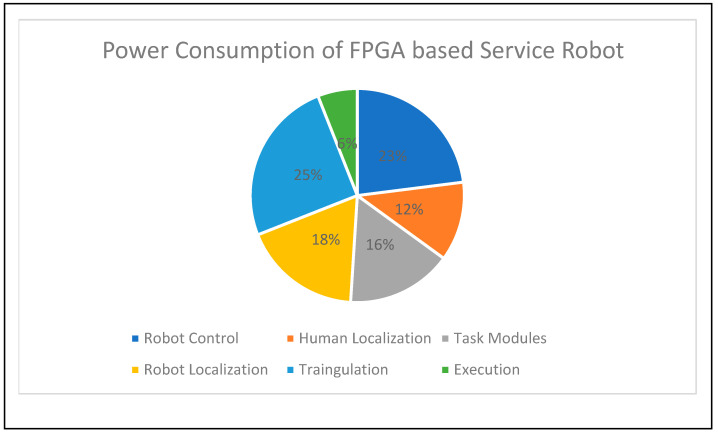
Device power consumption of service-based robot localization and navigation accelerator.

**Figure 13 sensors-24-06986-f013:**
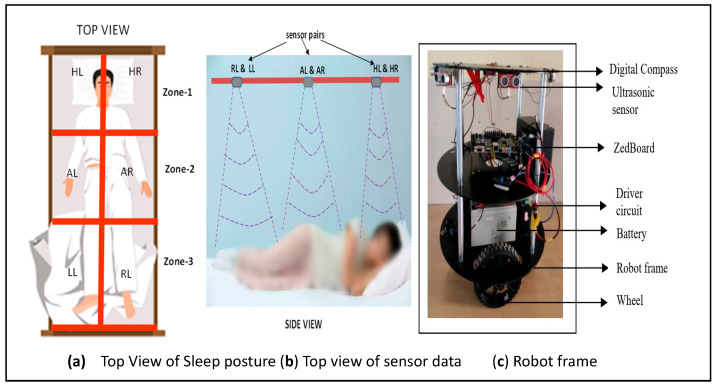
Human and robot interaction experimental setup.

**Figure 14 sensors-24-06986-f014:**
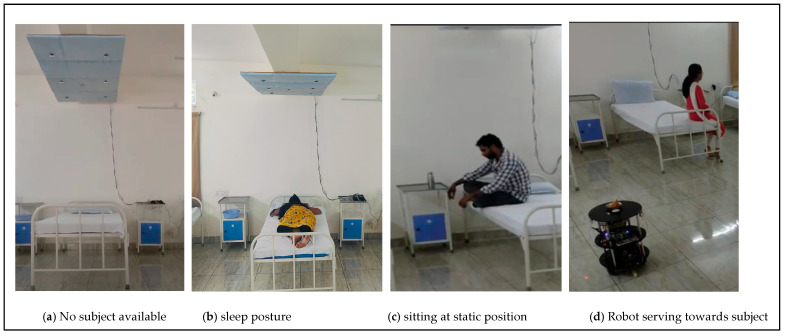
Real-time experimental setup of human and robot interaction.

**Figure 15 sensors-24-06986-f015:**
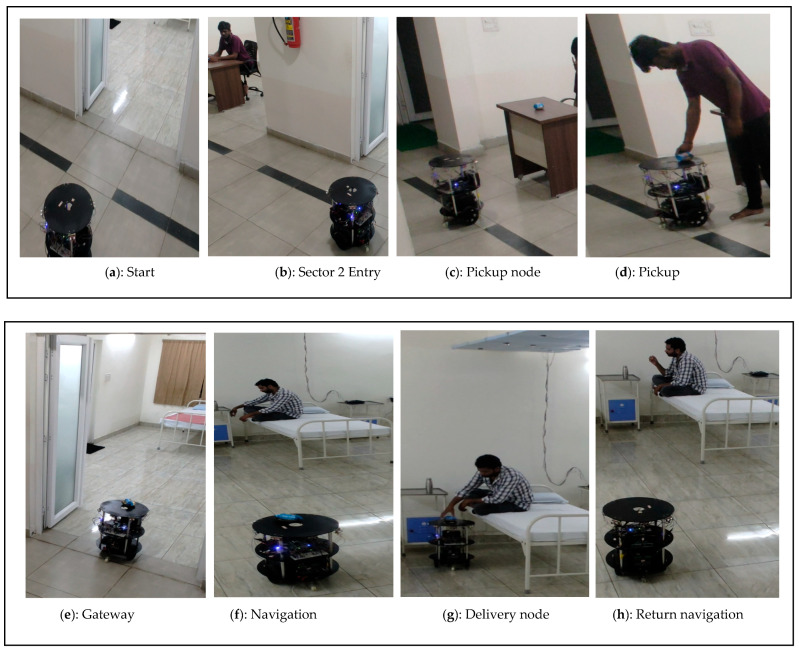
(**a**–**d**) Demonstration results of robot navigation from parking to pick up. (**e**–**h**) Demonstration results of human and robot interaction at static position.

**Figure 16 sensors-24-06986-f016:**
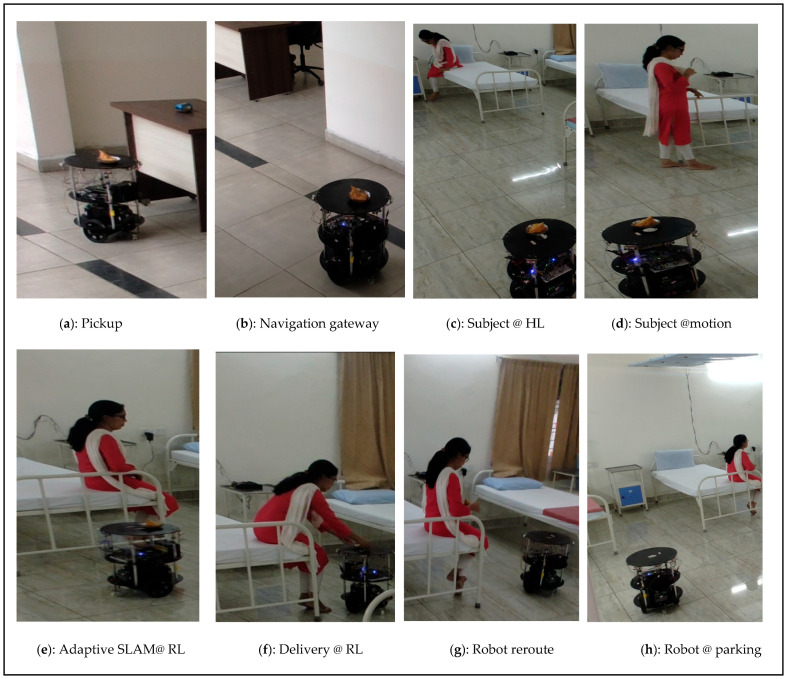
(**a**–**h**) Demonstration results of human and robot interaction at adaptive position.

**Figure 17 sensors-24-06986-f017:**
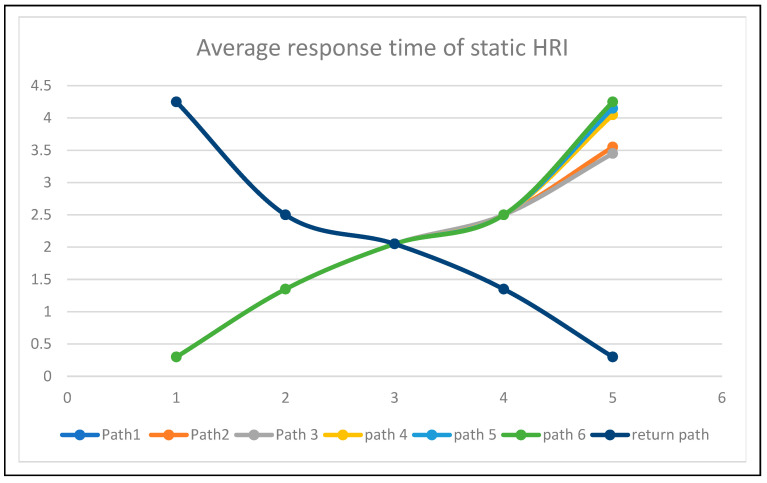
Average response time of human–robot interaction at static conditions.

**Figure 18 sensors-24-06986-f018:**
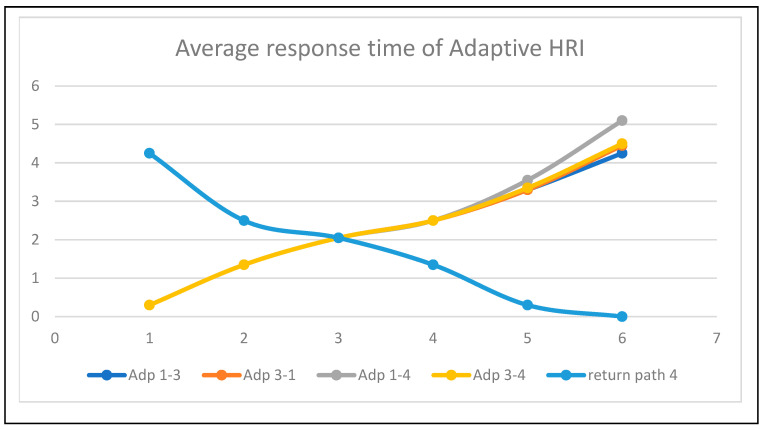
Average response time of human–robot interaction at adaptive conditions.

**Table 1 sensors-24-06986-t001:** Proposed research-related abbreviations.

Symbol	Abbreviation
S_PR_	S: ultrasonic sensors, as {H_R, H_L, A_R, A_L, R_L & L_L} P: Position of sensor at Head (H), Abdomen (A), Limb(L) R: Position at sides as Right ^®^, left(L)
PP	Past Position
CP	Current Position
S_R_	Service Robot
Dest_ Node	Destination Node

**Table 2 sensors-24-06986-t002:** Sensor fusion data for human localization.

Subject Seated at	Head_Right (HR)	Head_Left (HL)	Abdomen_Right (AR)	Abdomen_Left (AL)	Right Lower Limb_(RL)	Left Lower Limb_(LL)
Top position of right	1	0	0	0	0	0
Top position of left	0	1	0	0	0	0
Middle position of right	0	0	1	0	0	0
Middle position of left	0	0	0	1	0	0
Lower position of right	0	0	0	0	1	0
Lower position of left	0	0	0	0	0	1
Aliasing top of right	1	0	1	0	0	0
Aliasing top of head	1	1	0	0	0	0
Aliasing top of left	0	1	0	1	0	0
Aliasing middle of right	X	0	1	0	X	0
Aliasing middle of left	0	X	0	1	0	X
Aliasing lower of right	0	0	1	0	1	0
Aliasing lower of limbs	0	0	0	0	1	1
Aliasing lower of left	0	0	0	1	0	1

**Table 3 sensors-24-06986-t003:** FPGA resource utilization for human localization accelerator on Zed board.

Module	LUT	BRAM	DSP Slice
External modules and pose control unit	5246	14	22
Sleep posture PE	2714	06	08
Position binary classifier PE	1846	06	10
Adaptive sitting position PE	6330	18	18
Static and aliasing sitting position PE	4702	08	16
Execution modules, UART, and display	4342	6	08
Total	25180	58	82

**Table 4 sensors-24-06986-t004:** FPGA resource utilization for robot localization and navigation accelerator on Zed board.

Module	LUT	BRAM	DSP Slice
External modules and robot control unit	5246	14	22
Human localization PE	4404	10	12
Task assignment and sort PE	3916	12	10
Robot localization PE	5140	08	18
Triangulation navigation PE	7268	14	16
Execution modules, UART, and display	4342	6	08
Total	30,316	64	86

**Table 5 sensors-24-06986-t005:** Comparison of human and robot interaction with relevant research methods.

Reference Papers	Sensory Approach		Algorithm	Hardware	Number of Postures/Position	Pros	Accuracy	Cons
	Method	Fusion						
Q. Hu et al. 2021 [20]	1024 sensors Pressure sensor	Yes	HOG, SVM, and CNN	Arduino Nano and CPU	6	<400 ms, sampling and processing	86.94% to 91.24%	Contact approach
Matar et al. 2020 [37]	1728 FSR sensors	Yes	HOG + LBP, FFANN	CPU	4	Health monitoring	97%	More usage of sensors
R. Tapwal et al. 2023 [38]	Two flex force sensors	Yes	K-means	Arduino Uno and CPU	4	Health monitoring	~99.3%	consumes 17.5 W, contact approach
Hu, D et al. 2024 [39]	32 Piezoelectric sensor	Yes	S^3^CNN	N/A	4	Effectively detects nuanced pressure disturbances	93.0%	not applicable
Y. Tanaka et al. 2020 [40]	Camera	No	Amygdala	FPGA	_	Interaction with subject based face recognition	>90%	not applicable
T.Kim et al. 2024 [41]	Multi sensors and Camera	Yes	DFS, 3D routes, Vision algorithms	Intel NUC and Nvidia Jetson Xavier		Multi floor service and mapping	N/A	Costlier in implementation
Proposed	10 Ultrasonic sensors	Yes	Human Localization, Triangulation Navigation algorithm	FPGA		Parallel computing, <200 ns, sampling, and computation. Adaptive localization-based robot services	98.4%	PR flow will be preferred in future usage

## Data Availability

Data are contained within the article.

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
