# Peer review of "Adaptive FPGA-Based Accelerators for Human–Robot Interaction in Indoor Environments"

_sensors, 2024, doi:10.3390/s24216986_

Round 1
Reviewer 1 Report
Comments and Suggestions for Authors
see attachment.

Author Response
Thank you for your kind revision, it is helpful to us for enhance the quality of manuscript.

Reviewer 2 Report
Comments and Suggestions for Authors
The paper covers FPGA implementation, SLAM, human posture recognition, and healthcare but ends up not diving deep enough into any of these topics.
The authors present a broad introduction with a number of reference works regarding the existing research on sleep monitoring systems. Although interesting, this literature review is not directly relevant to FPGA-based accelerators. The key novelty (FPGA accelerators) suffers from the lack of clear examples or comparisons with state-of-the-art systems, thus diminishing this work's contribution.
Throughout the paper, terms like "human localization," "adaptive SLAM," and "triangulation-based navigation" are not always used consistently. The authors should define key terms early on and maintain consistency. The hardware details are complete and relevant for this work but the authors do not provide context about real-world applications, system benefits, or comparative advantages.
Some acronyms (such as SLAM) must be defined at the beginning of the paper.
Proofreading for both technical and linguistic clarity is necessary because there are grammatical issues and awkward sentence structures. For example, “timely decision-making is critical” is followed by disconnected technical descriptions that make it hard to follow by the reader.
I recommend reworking the introduction to clarify the problem and restructure the methodology to improve readability. Adding more practical context and robust validation of the system proposed by the authors will strengthen the overall contribution.
Comments on the Quality of English LanguageProofreading for both technical and linguistic clarity is necessary because there are grammatical issues and awkward sentence structures. For example, “timely decision-making is critical” is followed by disconnected technical descriptions that make it hard to follow by the reader.
Author Response
Thank you for your kind review, it is helpful for us to improve the quality of manuscript.
